# IllumiNeRF: 3D Relighting Without Inverse Rendering

**Xiaoming Zhao**[1,3*]   **Pratul P. Srinivasan**[2]   **Dor Verbin**[2]
**Keunhong Park**[1]   **Ricardo Martin-Brualla**[1]   **Philipp Henzler**[1]
[1]Google Research   [2]Google DeepMind   [3]University of Illinois Urbana-Champaign

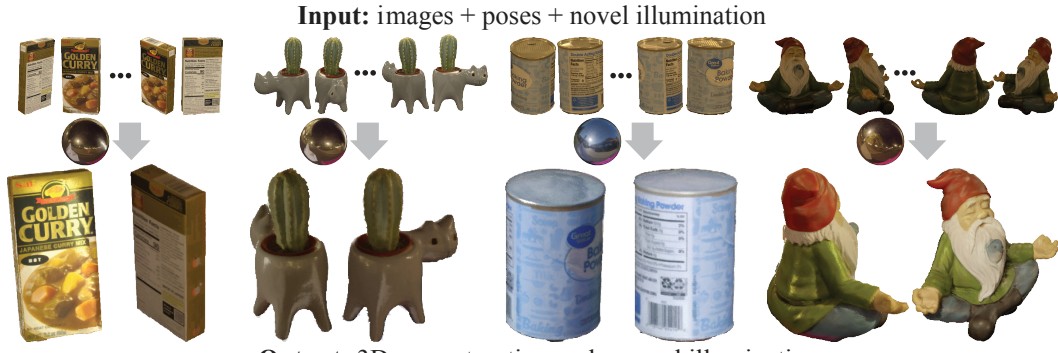

**Input:** images + poses + novel illumination

**Output:** 3D reconstruction under novel illumination

Figure 1: Given a set of posed input images under an *unknown* lighting (four exemplar images from the set are shown on top), IllumiNeRF produces high-quality novel views (bottom) relit under a target lighting (illustrated as chrome balls). Inputs obtained from the Stanford-ORB dataset [27].

## Abstract

Existing methods for relightable view synthesis — using a set of images of an object under unknown lighting to recover a 3D representation that can be rendered from novel viewpoints under a target illumination — are based on inverse rendering, and attempt to disentangle the object geometry, materials, and lighting that explain the input images. Furthermore, this typically involves optimization through differentiable Monte Carlo rendering, which is brittle and computationally-expensive. In this work, we propose a simpler approach: we first relight each input image using an image diffusion model conditioned on target environment lighting and estimated object geometry. We then reconstruct a Neural Radiance Field (NeRF) with these relit images, from which we render novel views under the target lighting. We demonstrate that this strategy is surprisingly competitive and achieves state-of-the-art results on multiple relighting benchmarks. Please see our project page at `illuminerf.github.io`.

## 1   Introduction

Capturing an object's appearance so that it can be accurately rendered in novel environments is a central problem in computer vision whose solution would democratize 3D content creation for augmented and virtual reality, photography, filmmaking, and game development. Recent advances in view synthesis [36] have made impressive progress in reconstructing a 3D representation that can be rendered from novel viewpoints, using just a set of observed images. However, those methods

---

[*]Work done as an intern at Google.

38th Conference on Neural Information Processing Systems (NeurIPS 2024).

typically only recover the appearance of the object under the captured illumination, and *relightable view synthesis* — rendering novel views of the captured object under arbitrary target environments — remains challenging.

Recent methods for recovering relightable 3D representations treat this task as *inverse rendering*, and attempt to estimate the geometry, materials, and illumination that jointly explain the input images using physically-based rendering methods. These approaches typically involve gradient-based optimization through differentiable Monte Carlo rendering procedures, which are noisy and computationally-expensive. Moreover, the inverse rendering optimization problem is brittle and inherently ambiguous; many potential sets of geometry, materials, and lighting can explain the input images, but many of these incorrect explanations produce obviously implausible renderings when rendered under novel unobserved illumination.

We propose a different approach that avoids inverse rendering and instead leverages a generative image model fine-tuned for the task of relighting. Given a set of images viewing an object and a desired target illumination, we use a single-image 2D Relighting Diffusion Model that outputs relit images of the object under the target illumination. Due to the ambiguous nature of the problem, each sample of the generative model encodes a different explanation of the object's materials, geometry and the input illumination. However, as opposed to optimization-based inverse rendering, such samples are all *plausible* relit images since they are the output of the trained diffusion model.

Instead of attempting to recover a single explanation of the underlying object's appearance, we sample multiple plausible relit images for each observed viewpoint, and treat the underlying explanations as samples of unobserved latent variables. To recover a final consistent 3D representation of the relit object, we use the full set of sampled relit images from all viewpoints to train a "latent NeRF" that reconciles all the samples into a single 3D representation, which can be rendered to produce plausible relit images from novel viewpoints.

The key contribution of our work is a new paradigm for relightable 3D reconstruction that replaces 3D inverse rendering with: generating samples with a single-image 2D Relighting Diffusion Model followed by distilling these samples into a 3D latent NeRF representation. We demonstrate that this strategy is surprisingly competitive and outperforms existing most 3D inverse rendering baselines on the TensoIR [23] and Stanford-ORB [27] relighting and view synthesis benchmarks.

## 2 Related Work

Our work addresses the task of relightable 3D reconstruction by using a lighting-conditioned diffusion model as a generative prior for single-image relighting. It is closely related to prior work in relightable 3D reconstruction, inverse rendering, and single-image relighting. Below, we review these lines of work and discuss how they relate to our proposed approach.

**Relightable 3D Reconstruction** The goal of relightable 3D reconstruction is to reconstruct a 3D representation of an object that can be relit by novel illumination conditions and rendered from novel camera poses. In scenarios where an object is observed under multiple lighting conditions [12], it is trivial to render its appearance under novel illumination that is a linear combination of the observed lighting conditions, due to the linear behavior of light. This approach is generally limited to laboratory capture scenarios where it is possible to observe an object under a lighting basis.

In more casual capture scenarios, the object is observed under just a single or a small handful of lighting conditions. Existing works typically address this setting using methods based on inverse rendering that explicitly factor an object's appearance into the underlying 3D geometry, object material properties, and lighting that jointly explain the observed images. State-of-the-art approaches to 3D inverse rendering [9, 10, 17, 23, 26, 33, 38, 46, 47] generally utilize the following strategy: they start with a neural field representation of 3D geometry (typically volume density as in NeRF [36], hybrid volume-surface representations as in NeuS [57] and VolSDF [59], or meshes extracted from neural field representations) from the input images, equip the model with a representation of surface materials (*e.g.* spatially-varying BRDF parameters) and lighting, and jointly optimize these factors through a differentiable physics-based rendering procedure [40]. While methods may differ in their choice of geometry, material, and lighting representations, and employ different techniques to accelerate the evaluation of the rendering integral, they generally all follow this same high-level inverse rendering strategy. Unfortunately, even if the geometry is known, inverse rendering

is a notoriously ambiguous problem [43, 52] and many combinations of materials and lighting can explain an object's appearance. However, not all of these combinations are plausible, and incorrect factorizations that explain observed images under one lighting condition may produce glaring artifacts when rendered under different lighting. Furthermore, differentiable physics-based rendering is computationally-expensive as thousands of samples are needed for Monte Carlo estimates of the rendering integral, typically requires custom implementations [2, 3, 22, 28, 32, 35, 54], and the resulting inverse rendering loss landscape is non-smooth and difficult to optimize effectively with gradient descent [14].

**Single Image Relighting**    Instead of using inverse rendering to recover object material parameters which can be relit with physically-based rendering techniques, we train a diffusion model that can directly sample from the distribution of relit images conditioned on a target lighting condition. This diffusion model is essentially a generative single-image relighting model. Early single image relighting techniques employed optimization-based inverse rendering [4]. Subsequent methods trained deep convolutional neural networks to output image geometry, materials, and lighting [29, 30], or in some cases, to directly output relit images [48, 7, 8].

Most related to our method are a few recent works that have trained diffusion models for single image relighting. LightIt [25] trains a model similar to ControlNet [63] to relight outdoor images under arbitrary sun positions conditioned on input normals and shading. DiffusionLight [41] estimates the lighting of an image by using a ControlNet to inpaint the color pixels of a chrome ball in the middle of the scene, from which an environment map can be recovered.

Most similar to our work is the concurrent method of DiLightNet [61] that focuses on single image relighting. DiLightNet uses a ControlNet-based [63] approach to condition a single-image relighting diffusion model on a target environment map. DiLightNet uses a set of "radiance cues" [15] — renderings of the object's geometry (obtained from an off-the-shelf monocular depth network) with various roughness levels under the target environment illumination — as conditioning. Our method instead focuses on 3D relighting, where multiple of images of an object are available. It uses a similar single-image relighting diffusion model conditioned on radiance cues. Unlike DiLightNet which uses geometry from monocular depth estimation to render radiance cues, we use geometry estimated from the input views using a state-of-the-art surface reconstruction method [56]. This allows our model to better model complex light transport effects such as interreflections caused by occluded geometry.

## 3    Method

### 3.1    Problem Formulation

Given a dataset of images of an object and corresponding camera poses $\mathcal{D} = \{(I_i, \pi_i)\}_{i=1}^N$, the general goal of relightable 3D reconstruction is to estimate a model with parameters $\theta$ that when rendered, produces relit versions of the dataset under unobserved target illumination $L^T$. This can be expressed as:

$$\theta^\star = \underset{\theta}{\arg\max}\, p(\mathcal{D}_\theta^T | \mathcal{D}), \tag{1}$$

where $\mathcal{D}_\theta^T \triangleq \left\{ \left(\text{relight}(\mathcal{D}, L^T, \pi_i, \theta), \pi_i\right) \right\}_{i=1}^N$ is a relit version of the original dataset under target illumination $L^T$ using model $\theta$. Note that Eq. (1) only maximizes the likelihood of the original given poses after relighting. However, by using view synthesis, we can then turn the collection of relit images into a 3D representation which can be rendered from arbitrary poses. For brevity, we therefore omit the implicit dependence of $\mathcal{D}^T$ in $\theta$.

This relighting problem has traditionally been solved by using inverse rendering. Inverse rendering techniques do not maximize the probability of the relit renderings, but instead recover a single point estimate of the most likely scene geometry $G$, materials $M$, and lighting $L$ (note that this is the "source" lighting condition for the observed images) that together explain the input dataset, and then use physically-based rendering to relight this factorized explanation under the target lighting. Inverse rendering seeks to recover $\theta^{\text{IR}} = (G^\star, M^\star)$, where:

$$G^\star, M^\star, L^\star = \underset{G,M,L}{\arg\max}\, p(G, M, L | \mathcal{D}) = \underset{G,M,L}{\arg\max}\, p(\mathcal{D} | G, M, L) p(G, M, L). \tag{2}$$

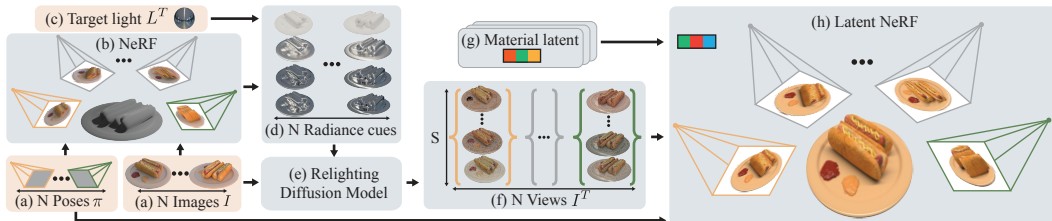

Figure 2: **Overview.** Given a set of images $I$ and camera poses $\pi$ in (a), we run NeRF to extract the 3D geometry as in (b). Based on this geometry and a target light shown in (c), we create radiance cues for each given input view as in (d). Next, we independently relight each input image using a single-image Relighting Diffusion Model illustrated in (e) and sample $S$ possible solutions for each given view displayed in (f). Finally, we distill the relit set of images into a 3D representation through a Latent NeRF optimization as in (g) and (h).

The first data likelihood term is computed by physics-based rendering of the estimated model and the second prior term is often factorized into separate handcrafted priors on geometry, materials, and lighting [23, 33, 43].

A relighting approach based on inverse rendering then renders each image $I$ in $\mathcal{D}$ corresponding to camera pose $\pi$ using the recovered geometry and materials, illuminated by the target lighting $L^T$, resulting in $\mathrm{relight}(\mathcal{D}, L^T, \pi, \theta^{\mathrm{IR}})$. This approach has three main issues. First, the differentiable rendering procedures used to compute the gradient of the likelihood term are computationally-expensive. Second, it requires careful modeling of light transport which is cumbersome and existing differentiable renderers do not account for many types of lighting and material effects seen in the real world. Third, there are often ambiguities between $M$ and $L$, meaning that any errors in their decomposition may be apparent in the relit data. It is quite difficult to design effective handcrafted priors on geometry, materials, and lighting, so inverse rendering procedures frequently recover explanations that have a high data likelihood (are able to render the observed data) but produce clearly incorrect results when re-rendered under different illumination.

## 3.2 Model Overview

We propose an approach that attempts to maximize the probability of relit images in Eq. (1) without using an explicit physically-based model of the object's lighting or materials. First, let us introduce a latent variable $Z$ that can be thought of as implicitly representing the input images' lighting along with the object's material and geometry parameters. We can write the likelihood of the relit data as:

$$p(\mathcal{D}^T|\mathcal{D}) = \int p(\mathcal{D}^T, Z|\mathcal{D})dZ = \int p(\mathcal{D}^T|Z,\mathcal{D})p(Z|\mathcal{D})dZ. \tag{3}$$

Introducing these latent variables lets us consider all relit renderings in the dataset, $\mathcal{D}_i^T \triangleq (I_i^T, \pi_i)$, as conditionally independent, since the rendering under the target lighting $L^T$ is deterministic given the object's geometry and materials. This enables writing the likelihood as:

$$p(\mathcal{D}^T|\mathcal{D}) = \int \underbrace{\left[\prod_{i=1}^{N} p(\mathcal{D}_i^T|Z_i,\mathcal{D}_i)\right]}_{\text{latent NeRF}} \underbrace{p(Z|\mathcal{D})}_{\text{latent prior}} dZ. \tag{4}$$

We propose to model this with a latent NeRF model, as used by Martin-Brualla *et al*. [34] that is able to render novel views under the target illumination for any sampled latent vector. We describe this model in Sec. 3.3. We train this NeRF model by generating a large quantity of sampled relit images with the same target lighting but with different (unknown) latent vectors using a *Relighting Diffusion Model* which we will describe in Sec. 3.4. In this way, the latent NeRF model effectively distills a large dataset of relit images sampled by the diffusion model into a single 3D representation that can render novel views of the object under the target lighting for any sampled latent.

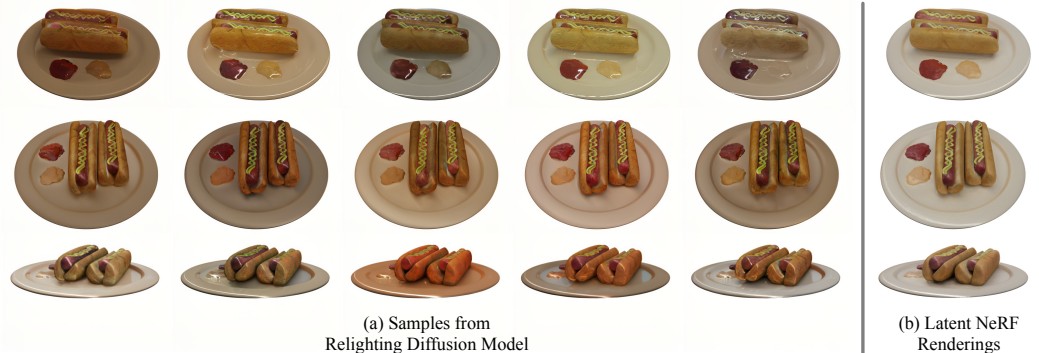

<table>
<tr><td align="center">(a) Samples from
Relighting Diffusion Model</td><td align="center">(b) Latent NeRF
Renderings</td></tr>
</table>

Figure 3: **Relit samples *vs*. latent NeRF.** (a) Samples of the Relighting Diffusion Model (Sec. 3.4) for the same target environment map, and (b) renderings from the optimized Latent NeRF (Sec. 3.3) for a fixed value of the latent. The diffusion samples correspond to different latent explanations of the scene and our latent NeRF optimization is able to effectively optimize these latent variables along with the NeRF model's parameters to produce consistent renderings for each latent explanation.

## 3.3 Latent NeRF Model

We wish to model the distribution in Eq. (4) in a manner that lets us render images that correspond to relit views of the object for any sampled latent $Z$. We choose to model this with a latent code NeRF 3D representation, inspired by prior works that condition NeRFs on latent codes to represent sources of variation such as the time of day during capture [34]. This latent NeRF optimizes a set of latent codes that are used to condition the view-dependent color function represented by the NeRF, enabling it to render novel views of the relit object under the target illumination for any sampled latent code. In our implementation, the latent NeRF's geometry does not depend on the latent code, so the latent code may be interpreted as only representing the object's material properties.

To optimize the parameters $\theta$ of the latent NeRF model, we maximize the log-likelihood, which by using Eq. (4), can be written as the following maximization problem:

$$\theta^{\star} = \underset{\theta}{\arg\max} \log p(\mathcal{D}_{\theta}^{T}|\mathcal{D}) = \underset{\theta}{\arg\max} \log \int \left[ \prod_{i=1}^{N} p(\mathcal{D}_{i}^{T}|Z_{i}, \mathcal{D}_{i}) \right] p(Z|\mathcal{D}) dZ. \quad (5)$$

Because integrating over all possible latents $Z$ is intractable, we use a heuristic inference strategy and replace the integral with the maximum a posteriori (MAP) estimate of $Z$:

$$\theta^{\star} \approx \underset{\theta}{\arg\max} \underset{Z}{\max} \left\{ \sum_{i=1}^{N} \log p(\mathcal{D}_{i}^{T}|Z_{i}, \mathcal{D}_{i}) + \log p(Z|\mathcal{D}) \right\}. \quad (6)$$

By assuming a Gaussian model over the data given the materials, the first term in Eq. (6) is a reconstruction loss over the images. However, since we do not have access to the true latent vector $Z$, we assume a uniform prior over them, turning the second term in Eq. (6) into a constant. In practice, similar to prior work on NeRFs optimized to generate new views given a dataset containing images with varying appearance, we rely on the NeRF model to resolve any mismatches in the appearance of different images [34]. See Fig. 3 for illustrations. The minimization of the negative log-likelihood can then be written as:

$$\theta^{\star} = \underset{\theta}{\arg\min} \underset{Z}{\min} \sum_{i=1}^{N} \|\mathcal{D}_{i}^{T} - \text{latent-NeRF}(\theta, Z_{i}, \pi_{i})\|^{2}. \quad (7)$$

## 3.4 Relighting Diffusion Model

In order to train the latent NeRF model described in Sec. 3.3, we use a Relighting Diffusion Model (RDM) to generate $S$ samples for each viewpoint from $p(\mathcal{D}_{i}^{T}|\mathcal{D}_{i})$. In other words, given an input image and target lighting $L^{T}$, the single-image RDM samples $S$ images corresponding to relit

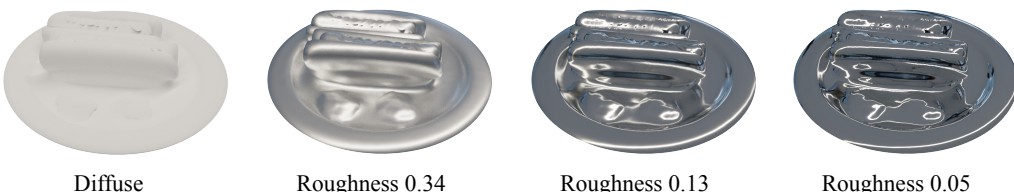

| Diffuse | Roughness 0.34 | Roughness 0.13 | Roughness 0.05 |

Figure 4: Example radiance cues for a view of the 'hotdog' scene.

versions of $D_i$ that have a high likelihood given the new target light $L^T$. We then associate each sample $s \in \{1, \ldots, S\}$ with its own latent code $Z_{i,s}$ and sum over all samples when training the latent NeRF (Eq. (7)).

Our RDM is implemented as an image denoising diffusion model that is conditioned by the input image and target lighting. To encode the target lighting, we use image-space radiance cues [15, 44, 61], visualized in Fig. 4. These radiance cues are generated by using a simple shading model to render a handful of images of the object's estimated geometry under the target lighting. This procedure is designed to provide information about the effects of specularities, shadows, and global illumination, without requiring the diffusion network to learn these effects from scratch. In our experiments, we use four different pre-defined materials to render radiance cues: one diffuse material with a pure white albedo, and three purely-specular materials with roughness values $\{0.05, 0.13, 0.34\}$. We use GGX [55] as the shading model. For more details, please refer to Sec. A.2.

The RDM architecture consists of a pretrained latent image diffusion model, similar to StableDiffusion [45], and uses a ControlNet [63] based approach to condition on the radiance cues. Please refer to Sec. A.3 for more architecture details.

## 4 Experiments

### 4.1 Experimental Setup

**Relighting Dataset** We render objects from Objaverse [13] under varying poses and illuminations. For each object, we randomly sample 4 poses, and render each under 4 different lighting conditions. We represent the lighting as HDR environment maps, and randomly sample from a dataset of 509 environment maps from Polyhaven [60]. For more details, see Sec. A.4.

**Evaluation Datasets** We evaluate our method on two datasets: TensoIR [23], a synthetic benchmark, and Stanford-ORB [27], a real-world benchmark. TensoIR contains renderings of four synthetic objects rendered under six lighting conditions. Following [23], we use the training split of 100 renderings with "sunset" lighting as input $\{I_i\}$. We then evaluate on 200 poses, each of which has renderings under five different environment maps, *i.e.*, "bridge", "city", "fireplace", "forest", and "night", for a total of 4000 renderings. Stanford-ORB is a real-world benchmark for inverse rendering on data captured in the wild. It contains 14 objects with various materials and captures each object under three different lighting settings, resulting in 42 (object, lighting) pairs. For the task of relighting, we are given images of an object under a single lighting condition and follow the benchmark protocol to evaluate relit images of the object under the two target lighting settings.

**Baselines** We compare our method to several existing inverse rendering approaches. On both benchmarks, we compare to NeRFactor [65] and InvRender [66]. On the synthetic benchmark, we additionally compare to TensoIR [23], the current top-performing approach on that benchmark. For the Stanford-ORB benchmark, we additionally compare to PhySG [62], NVDiffRec [38], NeRD [10], NVDiffRecMC [17], and Neural-PBIR [47].

**Our Model Inference** At inference time, the ideal embedding vector $Z$ that best corresponds to the actual material is unknown. One approach to find this vector is to optimize $Z$ to match a subset of the test set images (as in [34]). However, to ensure a fair comparison, we avoid this optimization. Instead, we set $Z = 0$ for all views when rendering test images.

Table 1: **TensoIR benchmark [23].** We evaluate four objects. Each object has five target lightings, each of which is associated with 200 poses, resulting in evaluating 4000 renderings in total. Running time for baselines are copied from [23]. Our time is A (geometry optimization on GPU) + B (diffusion sampling on TPU) + C (latent NeRF optimization on GPU). Best and 2nd-best are highlighted.

| | PSNR↑ | SSIM↑ | LPIPS↓ | Wall-clock Time↓ | Device |
|---|---|---|---|---|---|
| NeRFactor [65] | 23.383 | 0.908 | 0.131 | > 100 h | a RTX 2080 Ti |
| InvRender [66] | 23.973 | 0.901 | 0.101 | 15 h | a RTX 2080 Ti |
| TensoIR [23] | 28.580 | 0.944 | 0.081 | 5 h | a RTX 2080 Ti |
| Ours | 29.709 | 0.947 | 0.072 | 0.75 h + 1 h + 0.75 h | 16 A100 40GB + a TPUv5 |
| Ours (single GPU) | 29.245 | 0.946 | 0.073 | 2 h + 1 h + 2 h | a A100 40GB + a TPUv5 |

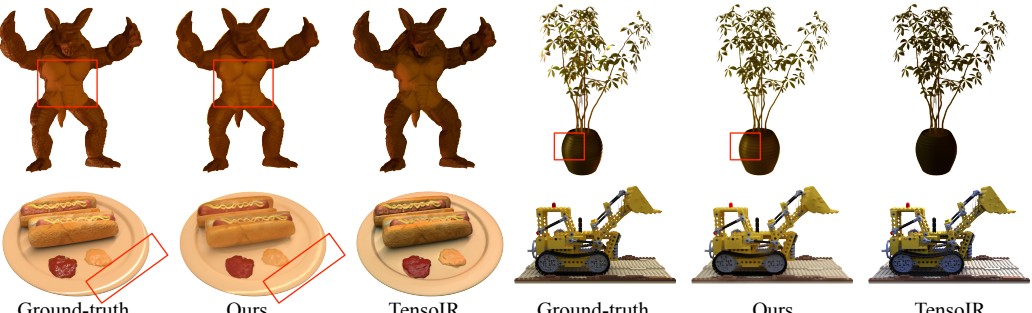

Figure 5: **Qualitative results on TensoIR.** Renderings from all approaches have been rescaled with respect to the ground-truth as mentioned in Eq. (4.1). Unlike TensoIR, our method faithfully recovers specular highlights and colors as indicated in red.

**Evaluation Metrics** For both benchmarks, we evaluate the quality of 3D relighting by reporting image metrics for rendered images. We report PSNR, SSIM [58], and LPIPS-VGG [64] on low dynamic range (LDR) images. Additionally, we report PSNR on high dynamic range (HDR) images on Stanford-ORB following the benchmark protocol, denoted as PSNR-H while the PSNR on LDR images is marked as PSNR-L. For approaches that do not produce HDR renderings, including ours, we convert the LDR renderings to linear values by using the inverse of the sRGB tone mapping curve. Due to the inherent ambiguities for the relighting task, we follow prior works [23, 27] and apply a channel-wise scale factor to RGB channels to match the ground truth image before computing metrics. Following established evaluation practices on Stanford-ORB, we compute the scale per output image individually whereas for TensoIR we compute a global scale factor that is used for all output images.[2]

## 4.2 Benchmarking

Unless otherwise specified, all results are produced using $S = 16$ samples (see Sec. 3.4) and make use of 16 A100 40GB GPUs (batch size of $2^{14}$ rays for NeRF optimization). We also provide results on a single A100 40GB GPU (batch size of $2^{13}$ for NeRF optimization).

We report quantitative results on the TensoIR benchmark in Tab. 1, and show qualitative examples in Fig. 5. We significantly outperform all competitors quantitatively on all metrics with comparable or improved wall-clock time. Visually our method is capable of recovering specular highlights whereas prior methods struggle to model these.

Similarly, we report results on Stanford-ORB in Tab. 2 and Fig. 6. Our proposed approach quantitatively improves upon all baselines, except those of Neural-PBIR [47], indicating the effectiveness of IllumiNeRF in real world scenarios. Note that although Neural-PBIR achieves better metrics than us, Fig. 6 shows that their relighting results are mostly diffuse, even for highly-glossy objects, and that they lack many of the strong specular highlights that our method is able to recover. This behavior of their model may explain their better metrics despite worse qualitative performance for specular highlights, because the illumination maps provided by Stanford-ORB do not correspond to the

---

[2]Please refer to https://github.com/StanfordORB/Stanford-ORB/blob/962ea6d2cc/scripts/test.py#L36 and https://github.com/Haian-Jin/TensoIR/blob/2a7a4d00/renderer.py#L12.

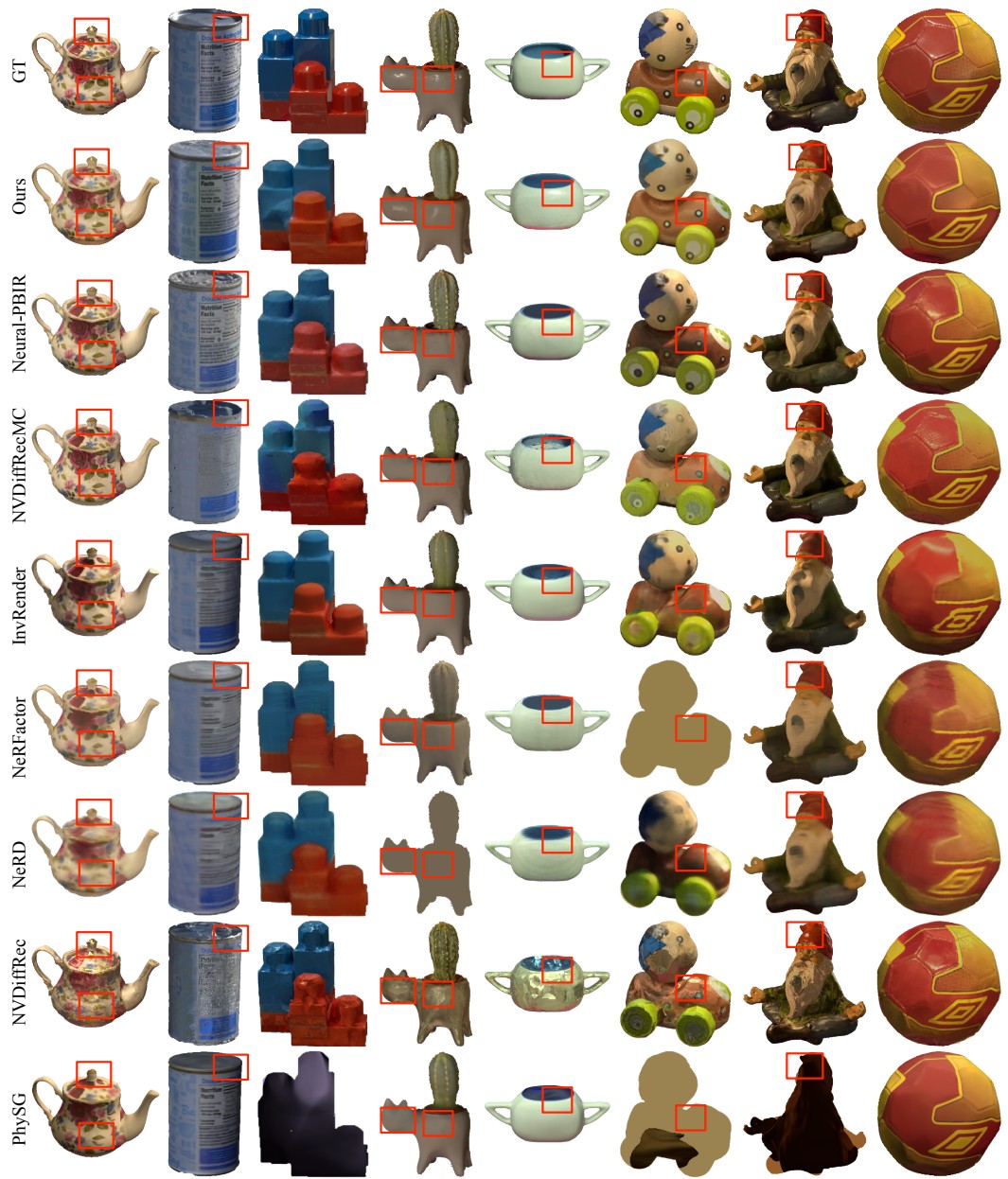

Figure 6: **Qualitative results on Stanford-ORB.** Renderings from all approaches have been rescaled with respect to the ground-truth as mentioned in Sec. 4.1. Areas where our approach performs well are highlighted. Our approach produces high-quality renderings with plausible specular reflections.

incident illumination at the object's location, since they were captured using a light probe which was moved for each image in the dataset [27]. This means that even given perfect materials and geometry, the images relit by *any* method cannot match with the true captured images, which is most noticeable in specular highlights. This mismatch penalizes methods like ours, which recover such specularities, over ones that recover mostly diffuse appearance with no apparent specular highlights [52]. For a more detailed discussion see Sec. B.

We also provide qualitative results for different latent codes in Fig. 7. These results demonstrate that the optimized latent codes effectively capture various plausible explanations of the materials.

Table 2: **Stanford-ORB benchmark [27]**. We evaluate 14 objects, each of which was captured under three different lightings. For each (object, lighting) pair, we evaluate renderings of the same object under the other two lightings, resulting in evaluating 836 renderings. †denotes models trained with the ground-truth 3D scans and pseudo materials optimized from light-box captures. Best and 2nd-best are highlighted.

| | PSNR-H↑ | PSNR-L↑ | SSIM↑ | LPIPS↓ |
|---|---|---|---|---|
| NVDiffRecMC [17]† | 25.08 | 32.28 | 0.974 | 0.027 |
| NVDiffRec [38]† | 24.93 | 32.42 | 0.975 | 0.027 |
| PhySG [62] | 21.81 | 28.11 | 0.960 | 0.055 |
| NVDiffRec [38] | 22.91 | 29.72 | 0.963 | 0.039 |
| NeRD [10] | 23.29 | 29.65 | 0.957 | 0.059 |
| NeRFactor [65] | 23.54 | 30.38 | 0.969 | 0.048 |
| InvRender [66] | 23.76 | 30.83 | 0.970 | 0.046 |
| NVDiffRecMC [17] | 24.43 | 31.60 | 0.972 | 0.036 |
| Neural-PBIR [47] | 26.01 | 33.26 | 0.979 | 0.023 |
| Ours | 25.42 | 32.62 | 0.976 | 0.027 |
| Ours (single GPU) | 25.56 | 32.74 | 0.976 | 0.027 |

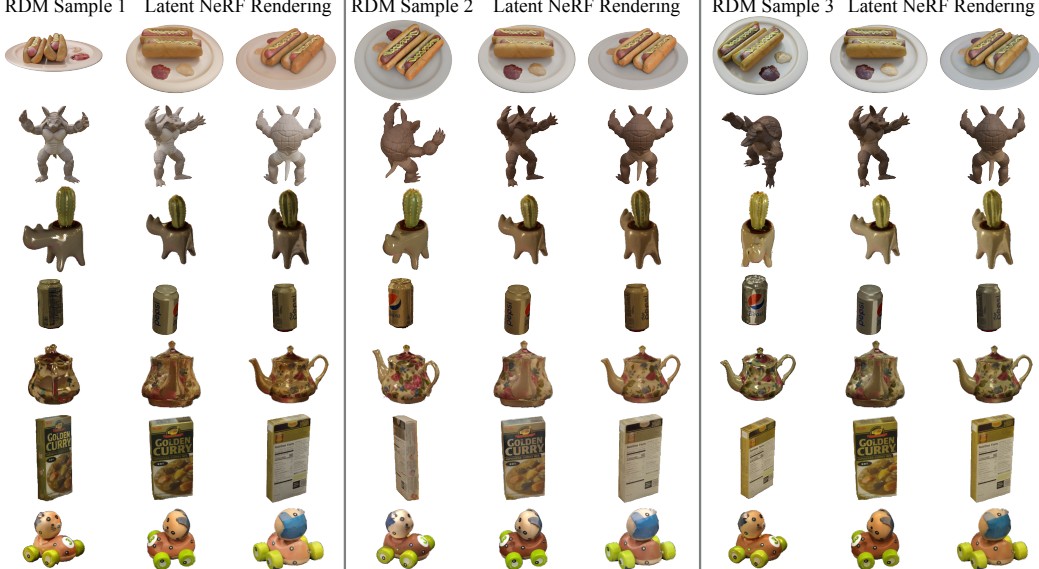

RDM Sample 1  Latent NeRF Rendering | RDM Sample 2  Latent NeRF Rendering | RDM Sample 3  Latent NeRF Rendering

Figure 7: **Renderings from various latents.** Each column shows 1) a Relighting Diffusion Model (RDM) sample and 2) two latent NeRF renderings using the sample's latent code. The diffusion samples are selected uniformly from all $N$ (#views) $\times$ $S$ (#samples per view) diffusion generations. Each row shows results from the same object and lighting with latent codes capturing various plausible explanations of the materials.

## 4.3 Ablations

We evaluate ablations of our model on TensoIR's hotdog scene in Tab. 3, and visualize them in Fig. 8. We reach the following conclusions: **1) The latent NeRF model is essential:** optimizing a standard NeRF cannot reconcile variations across views, even if we only generate a single sample per viewpoint for optimization ($S = 1$). **2) More diffusion samples help:** by increasing $S$, the number of samples from the RDM per viewpoint, we observe consistent improvements across almost all metrics. This corroborates our intuition that using an increased number of samples helps the latent NeRF effectively fit the target distribution (Eq. (4)) in a more stable way.

Table 3: **Ablations.** We conduct ablation studies on the Hotdog scene from TensoIR [23]. We evaluate renderings of 200 novel test camera poses, each under five target environment map lighting conditions, resulting in evaluating 1000 renderings in total. Best is highlighted.

| S | Latent | PSNR↑ | SSIM↑ | LPIPS↓ |
|---|--------|-------|-------|--------|
| 1 | ✗ | 24.957 | 0.921 | 0.099 |
| 1 | ✓ | 26.321 | 0.925 | 0.097 |
| 4 | ✓ | 27.409 | 0.936 | 0.087 |
| 16 | ✓ | 27.950 | 0.939 | 0.082 |

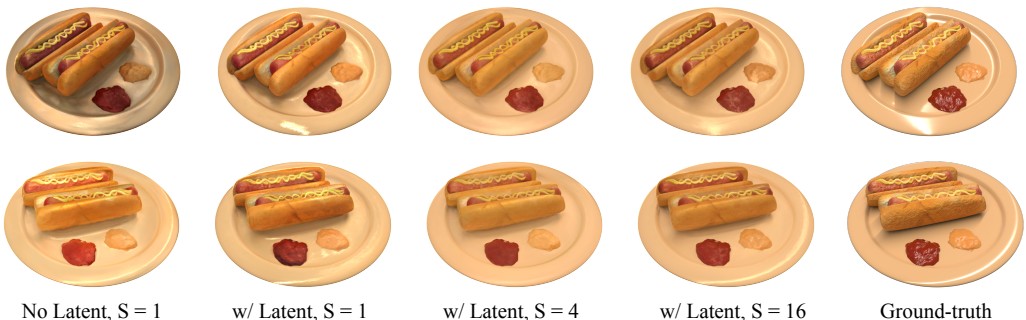

| No Latent, S = 1 | w/ Latent, S = 1 | w/ Latent, S = 4 | w/ Latent, S = 16 | Ground-truth |

Figure 8: Using a standard NeRF instead of a latent NeRF model is unable to reconcile training samples with different underlying latent explanations. Using a latent NeRF model significantly increases the accuracy of rendered specular appearance, and increasing the number of samples $S$ from the RDM used to train the latent NeRF model further increases the quality of the output renderings.

## 4.4 Limitations

Our model relies on high quality geometry estimated by UniSDF [56] (see Sec. A.1) to provide sufficiently good radiance cues for conditioning the RDM (Sec. 3.4). Any missing structure will lead our model to miss specular reflections, as seen on the top left of the salt can result in Fig. 6's second column. Errors in geometry also affect the quality of synthesized novel views, *e.g.* the missing thin branches from the plant in Fig. 5 or fine details of the cactus (column 4) in Fig. 6. Note that our RDM, trained on high-quality synthetic geometry, will inherently improve with future advances in geometry reconstruction. Our approach is not suited for real-time relighting, as it requires generating new samples with the RDM and optimizing a NeRF for any new target lighting condition.

## 5 Conclusion

We have proposed a new paradigm for the task of relightable 3D reconstruction. Instead of decomposing an object's appearance into lighting and material factors and then relighting the object with physically-based rendering, we use a single-image Relighting Diffusion Model (RDM) to sample a varied collection of proposed relit images given a target illumination, and distill these samples into a single consistent 3D latent NeRF representation. This 3D representation can be rendered to synthesize novel views of the object under the target lighting. Perhaps surprisingly, this paradigm consistently outperforms existing inverse rendering methods on synthetic and real-world object relighting benchmarks. This new paradigm's success is likely due to the RDM's ability to generate a large number of proposals for the new relit images. This is in contrast to prior works based on inverse rendering, which first estimates a single material model and then uses it for relighting, since errors in material estimation may propagate to the relit images. We believe that this paradigm may be used to improve data capture, material and lighting estimation, and that it may be used to do so robustly on real-world data.

## Acknowledgements

We would like to thank Ben Poole and Ruiqi Gao for insightful discussions. We thank Yunzhi Zhang and Zhengfei Kuang for providing their qualitative results for the Stanford-ORB [27] baseline, and Haian Jin for the TensoIR [23] baseline results. We are also grateful to Abhijit Kundu and Henna Nandwani for their infrastructure support.

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

# Appendix – IllumiNeRF: 3D Relighting Without Inverse Rendering

This appendix is organized as follows:

1. Sec. A provides more implementation details;

2. Sec. B details the inconsistent illumination issue on Stanford-ORB.

## A  Additional Implementation Details

### A.1  Latent NeRF Model and Geometry Estimator

We use JAX [11] to implement both the geometry estimator and Latent NeRF model as UniSDF [56], a state-of-the-art volume rendering approach based on a signed distance function (SDF). The advantage of using UniSDF is that it enables easily extracting a mesh from the SDF, which we can then import into a standard rendering engine such as Blender [50] in order to compute radiance cues. Additionally, UniSDF decouples geometry from appearance, allowing us to fix the weights related to geometry and only optimize for weights that model the appearance. Note that future NeRF/SDF approaches with improved geometric reconstruction can be seamlessly integrated in our method.

Our parameterization of the UniSDF model is similar to the one used in the original paper for the DTU dataset [1], with four key changes. First, we reduce the number of rounds of proposal sampling (as introduced by mip-NeRF 360 [5]) from two to one, using 64 proposal samples. Second, we use the asymmetric predicted normal loss from NeRF-Casting [53]:

$$\mathcal{L}_p = \sum_i \left( \lambda_1 \omega_i \| \cancel{\nabla} \mathbf{n}_i - \cancel{\nabla} \mathbf{n}_i' \|^2 + \lambda_2 \cancel{\nabla} \omega_i \| \mathbf{n}_i - \cancel{\nabla} \mathbf{n}_i' \|^2 + \lambda_3 \cancel{\nabla} \omega_i \| \cancel{\nabla} \mathbf{n}_i - \mathbf{n}_i' \|^2 \right), \qquad \text{(S1)}$$

where $\omega_i$ is the volume rendering weight of the $i$-th sample, $\cancel{\nabla}$ denotes the stop-gradient operator, $\mathbf{n}_i$ and $\mathbf{n}_i'$ are the $i$-th sample's density normals and predicted normals respectively (see [51]), and we set $\lambda_1 = \lambda_2 = 10^{-3}$, $\lambda_3 = 10^{-2}$. Third, like NeRF-Casting [53], we use an additional hash grid encoding [37] with 15 scales between a resolution of 32 and 4096, used only for outputting predicted normals. Fourth, we further encourage the local smoothness of the predicted normals $\mathbf{n}'$ by using a smoothness loss similar to [39, 65]:

$$\mathcal{L}_s = \lambda_4 \sum_i \omega_i \| \mathbf{n}'(\mathbf{x}_i + \varepsilon) - \mathbf{n}'(\mathbf{x}_i) \|^2, \qquad \text{(S2)}$$

where $\mathbf{x}_i$ is the 3D position of the $i$-th sample, and $\varepsilon \sim \mathcal{N}(0, \sigma^2 I)$ is an isotropic Gaussian random variable used to perturb the sample locations. We set $\lambda_4 = 0.1$ and $\sigma = 0.01$.

We find that these modifications result in better and smoother geometry necessary for our model's ability to relight objects with specular highlights.

Finally, to incorporate the GLO embeddings, we utilize an MLP to predict an element-wise scale and shift value to be applied to the 'bottleneck' feature of UniSDF, similar to AffineGLO in Zip-NeRF [6].

For both geometry estimation and latent NeRF optimization, we utilize the Adam [24] optimizer with $\beta_1 = 0.9$, $\beta_2 = 0.99$, and $\varepsilon = 1 \times 10^{-15}$. We decay our learning rate logarithmically from $5 \times 10^{-3}$ to $5 \times 10^{-4}$ over 25k training iterations with cosine-scheduled warmup in the first 500 steps.

### A.2  Radiance Cues

**Geometry**  To extract radiance cues we first optimize UniSDF [56] on the input images. After optimization, we convert the SDF representation to a mesh using marching cubes [31] with threshold set to be zero.

**Rendering**  We use Blender Cycles [50], a physically-based path-tracer to render the radiance cues. We run Blender via the Kubric python wrapper [16], and we use the estimated geometry with the predefined materials based on the GGX material model [55], as described in Sec. 3.4.

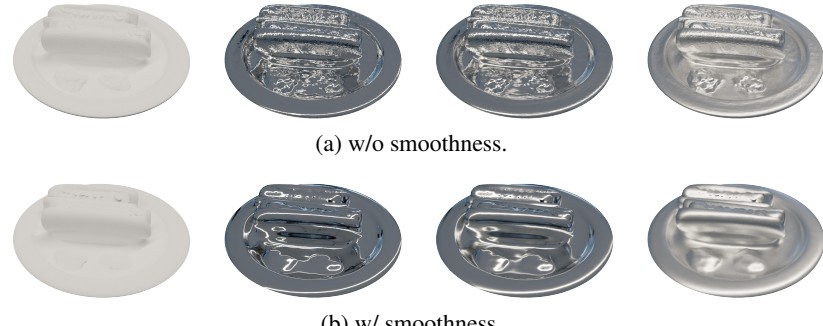

(a) w/o smoothness.

(b) w/ smoothness.

Figure S1: **Effects of shading normal smoothing function.**

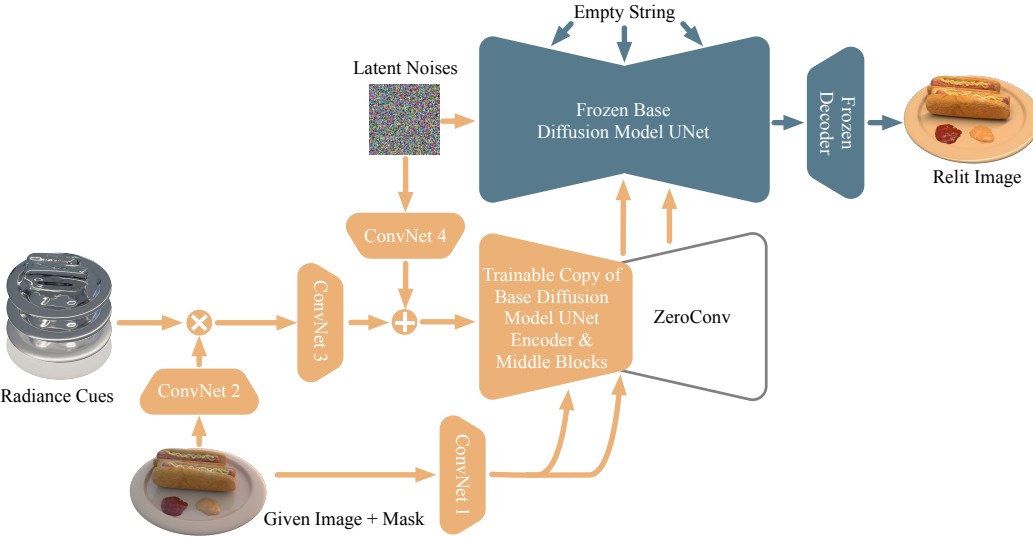

Figure S2: **Schematics of our ControlNet-based diffusion model.**

**Shading Normals** In order to produce smoothly-varying specular highlights which look realistic, we need the normals used for shading to be smooth. By default, Blender computes normals for shading based on the input geometry, which may be noisy. To mitigate this, we can feed the predicted normals $\mathbf{n}'$ described in Sec. A.1 to Blender and enable its shading normal smoothing function which applies to the predicted normals, and uses them for shading. However, over-smoothness may harm the photorealism of the rendered shadows. See Fig. S1 for qualitative comparison on radiance cues rendered without enabling the shading normal smoothing (Fig. S1a) and with the feature enabled (Fig. S1b). In our implementation, we exploit a hybrid strategy: we utilize radiance cues without smoothness for the diffuse material and use radiance cues with smoothness for the specular materials. Concretely, our final radiance cues are composed of the first rendering in Fig. S1a and the right three ones in Fig. S1b.

### A.3 Relighting Diffusion Model

We implement our relighting diffusion model in JAX [11]. We illustrate the architecture of the model for inference in Fig. S2. We build upon a text-to-image latent diffusion model which is similar to the model of Rombach *et al.* [45]. It denoises gaussian noise of size $64 \times 64 \times 8$ and decodes the output latent features into a relit image of size $512 \times 512 \times 3$. The model was not conditioned on text input, receiving only empty strings via a CLIP text encoder [42]. During training the base model is frozen.

Following ControlNet [63], we create a trainable copy of the base diffusion model's UNet encoder and middle blocks and append them with a ZeroConv-based blocks to the frozen base model. The

Table S1: **Fig. S2's ConvNet 1 Structure.** Convolution layer's definition is represented as (kernel size, stride, padding). We use SiLU [19] as the activation function between layers. Layer 8 uses zero initialization while the other layers use Flax's [18] default initialization[3]. In our implementation, we have $H = W = 512$.

| Index | Layer | Output Shape |
|---|---|---|
| 0 (input) | - | $H \times W \times 4$ |
| 1 | (3, 1, 1) | $H \times W \times 16$ |
| 2-1 | (3, 1, 1) | $H \times W \times 16$ |
| 2-2 | (3, 2, 1) | $H/2 \times W/2 \times 32$ |
| 3-1 | (3, 1, 1) | $H/2 \times W/2 \times 32$ |
| 3-2 | (3, 2, 1) | $H/4 \times W/4 \times 64$ |
| 4-1 | (3, 1, 1) | $H/4 \times W/4 \times 64$ |
| 4-2 | (3, 2, 1) | $H/8 \times W/8 \times 128$ |
| 5-1 | (3, 1, 1) | $H/8 \times W/8 \times 128$ |
| 5-2 | (3, 2, 1) | $H/16 \times W/16 \times 256$ |
| 6-1 | (3, 1, 1) | $H/16 \times W/16 \times 256$ |
| 6-2 | (3, 2, 1) | $H/32 \times W/32 \times 512$ |
| 7-1 | (3, 1, 1) | $H/32 \times W/32 \times 512$ |
| 7-2 | (3, 2, 1) | $H/64 \times W/64 \times 512$ |
| 8 | (3, 1, 1) | $H/64 \times W/64 \times 1024$ |
| 9 | flatten | $(H/64 \times W/64) \times 1024$ |

Table S2: **Fig. S2's ConvNet 3 Structure.** Convolution layer's definition is represented as (kernel size, stride, padding). We use SiLU [19] as the activation function between layers. Layer 5 uses zero initialization while the other layers uses Flax [18] default initialization[3]. In our implementation, we have $H = W = 512$.

| Index | Layer | Output Shape |
|---|---|---|
| 0 (input) | - | $H \times W \times 12$ |
| 1 | (3, 1, 1) | $H \times W \times 16$ |
| 2-1 | (3, 1, 1) | $H \times W \times 16$ |
| 2-2 | (3, 2, 1) | $H/2 \times W/2 \times 32$ |
| 3-1 | (3, 1, 1) | $H/2 \times W/2 \times 32$ |
| 3-2 | (3, 2, 1) | $H/4 \times W/4 \times 96$ |
| 4-1 | (3, 1, 1) | $H/4 \times W/4 \times 96$ |
| 4-2 | (3, 2, 1) | $H/8 \times W/8 \times 256$ |
| 5 | (3, 1, 1) | $H/8 \times W/8 \times 320$ |

given masked image and radiance cues are first fed through ConvNet 2 (see Fig. 4 in [61] for details) and ConvNet 3 (see Tab. S2). The resulting output is added to the output of the latent noise, which is fed through ConvNet 4. ConvNet 4 consists of a single convolution layer with kernel size 3, stride 1, padding 1, and 320 output channels. Given that the trainable copy was designed for tokenized text input, the masked image is first fed through ConvNet 1 (see Tab. S1) to generate representative embeddings. To ensure compatibility between the output of ConvNet 1 (size 64) and the CLIP [42] encoder's text output shape, zero-valued tensors are appended, increasing the size to 77.

We train the diffusion model using an approach similar to ControlNet [63], with a large dataset of synthetic objects rendered under multiple lighting conditions. Each training example for fine-tuning consists of a pair of images that view the same object with the same camera parameters, illuminated by two different environment map (see Sec. A.4). We fine-tune the diffusion model to predict one of these two images, given the other image as well as the corresponding radiance cues rendered using the synthetic object's geometry. Note that for synthetic objects, we do not need to estimate the geometry $G$ nor to enable the Blender normal smoothing function to compute the radiance cues since we already have the ground-truth meshes and the normals from synthetic objects are smooth enough. We fine-tune the base model for 150k steps using batch size of 512 examples and a learning rate of $10^{-4}$, which is linearly warmed up from 0 over the first 1k steps. The fine-tuning takes around 2 days on 32 TPUv5 chips. Besides, we always use the empty string as the text input to effectively make the fine-tuned model image-based.

At inference time, we use the DPPM scheduler [21] without classifier-free guidance [20] to produce samples at $512 \times 512$ resolution.

## A.4 Training Data Processing

We use Objaverse [13] as the synthetic dataset. To filter out low-quality objects, we use the list from [49] to get our initial set of 156,330 ones.[4] By additionally removing (semi-)transparent ones, we have a final set of 152,649 objects. If the object only contains geometry, we manually assign a homogeneous texture (ShaderNodeBsdfDiffuse) with a color uniformly sampled from $[0, 1]^3$. Further, if the object does not have the material information, we assign it a Blender Glossy BSDF

---

[3]https://github.com/google/flax/blob/144486b5fa7b3dfb/flax/core/nn/linear.py#L27
[4]https://github.com/ashawkey/objaverse_filter/tree/dc9e7cd0df8626f30df02bb

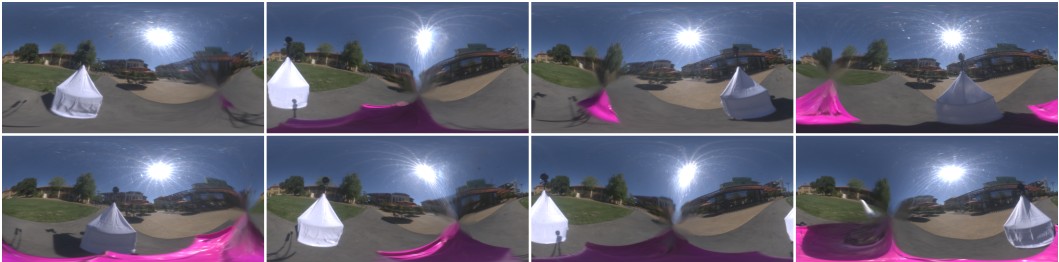

Figure S3: Stanford-ORB's per-image illuminations from `teapot_scene002`: inconsistent sun and tripod shape/location.

Table S3: **Issues on illuminations of Stanford-ORB [27]**. We create two sets of reference renderings with the ground-truth geometry and material: 1) using a **fixed** illumination to render all evaluation images for a same (object, lighting) pair; and 2) using the **per-image** illumination provided by the benchmark. We then evaluate each approach's renderings with respect to the two sets of reference renderings respectively. We also list each approach's illumination selection in the second column. For each row, better performance between the two evaluations is highlighted . Apparently and consistently, the numerical results favor matched illumination selection.

| | Illumination | Reference w/ **Fixed** Illumination | | | | Reference w/ **Per-image** Illumination | | | |
|---|---|---|---|---|---|---|---|---|---|
| | Selection | PSNR-H↑ | PSNR-L↑ | SSIM↑ | LPIPS↓ | PSNR-H↑ | PSNR-L↑ | SSIM↑ | LPIPS↓ |
| PhySG [62] | Per-image | 22.55 | 28.05 | 0.959 | 0.056 | 22.71 | 28.19 | 0.959 | 0.055 |
| NVDiffRec [38] | Per-image | 23.47 | 29.35 | 0.960 | 0.037 | 23.71 | 29.60 | 0.960 | 0.037 |
| NeRD [10] | Per-image | 24.05 | 30.20 | 0.968 | 0.053 | 24.22 | 30.36 | 0.969 | 0.053 |
| NeRFactor [65] | Per-image | 24.38 | 30.70 | 0.970 | 0.049 | 24.55 | 30.85 | 0.970 | 0.048 |
| InvRender [66] | Per-image | 24.50 | 30.75 | 0.970 | 0.047 | 24.68 | 30.93 | 0.971 | 0.046 |
| NVDiffRecMC [17] | Per-image | 25.17 | 31.19 | 0.970 | 0.037 | 25.45 | 31.53 | 0.970 | 0.036 |
| Ours | Fixed | 26.29 | 32.45 | 0.973 | 0.029 | 26.00 | 32.11 | 0.973 | 0.029 |
| Ours (single GPU) | Fixed | 26.34 | 32.53 | 0.974 | 0.029 | 26.05 | 32.17 | 0.973 | 0.029 |
| Real Images | – | 26.25 | 32.69 | 0.975 | 0.024 | 26.73 | 33.27 | 0.977 | 0.023 |

material (`ShaderNodeBsdfGlossy`), whose roughness value is uniformly sampled from $[0.02, 0.5]$ and base color is set to be the same as the homogeneous texture. The mixing factor between the specular and diffuse materials (`ShaderNodeMixShader`) is uniformaly sampled from $[0, 1]$.

As we discussed in Sec. A.3, our diffusion training requires image pairs under different lightings. For this, we select 509 equirectangular environment maps from [60]. For each object, we sample four camera poses on a sphere centered around it. For each camera, we randomly sample two environment maps and augment them with random horizontal shift, vertical flip, and RGB channel shuffle. We then use Blender's Cycle path tracer to render an image of resolution $512 \times 512$ with 512 samples per pixel for each environment map using a camera whose focal length is set to be 512.

# B  Stanford-ORB Illumination Issues

Stanford-ORB provides estimated **per-image** illumination via moving a light probe for each image, see Fig. S3 for an example. Ideally, fixed illumination per object would match reality, but aligning the object and light probe is challenging. This limitation in the Stanford-ORB benchmark can significantly affect results, especially in areas with specular highlights as demonstrated in Tab. S3. Consequently, there is no "correct" way to do relighting in Stanford-ORB: our results use fixed illumination, while competitors use per-image illumination.

We rendered the ground truth geometry and materials, obtained by Stanford-ORB in a controlled studio environment (see [27]'s Sec.3.2.2). Each view was rendered under **fixed** illumination (consistent environment map) and **per-image** illumination (unique environment map per view). Fig. S4 shows both renderings alongside the corresponding real image. Note the significant variations, especially in areas with specular highlights (see marked regions and PSNR).

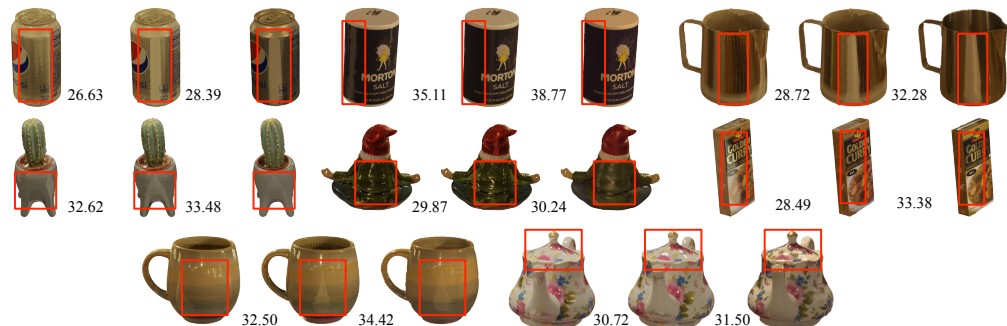

Figure S4: For each object, from left to right, we show 1) reference rendering w/ **fixed** illumination; 2) reference rendering w/ **per-image** illumination" (see Tab. S3 caption for details); and 3) the real captured image. We show PSNR-L between 1) *vs.* 3) and 2) *vs.* 3) respectively. Different illumination settings vary significantly.

We computed metrics for each method using both reference renderings as ground truth whose quantative results are in Tab. S3. Our method excels under fixed illumination but performs worse with per-image illumination. Competitors show the opposite trend, doing better with per-image illumination. Neural-PBIR did not release code or complete results, thus it is missing from the table.

