# OpenReview forum: "IllumiNeRF: 3D Relighting Without Inverse Rendering"
_NeurIPS.cc/2024/Conference — NeurIPS 2024 poster_

### Official Review · Reviewer_mZi1 · 2024-07-08

**Soundness:** 3
**Presentation:** 4
**Contribution:** 3
**Rating:** 7
**Confidence:** 4

**Summary:**

This paper introduces a new paradigm for the relightable 3D reconstruction: leverage a Relighting Diffusion Model to generate relit images and distill them into a latent NeRF. The proposed paradigm demonstrates superior performance compared to existing methods, which are based on inverse rendering, on both sythetic and real-world datasets.

**Strengths:**

- The writing is clear and the content is well-structured. The authors provide comprehensive implementation details.

- The paper addresses a long-standing challenge by introducing a novel paradigm, which diverges from conventional inverse rendering methods. This approach offers a promising alternative for relighting.

- Thorough numerical experiments showcase the effectiveness of the proposed method. The results are convincing.

**Weaknesses:**

One major weakness of this method is its efficiency. Unlike traditional methods that can directly relight a 3D scene once the inverse rendering is performed, the proposed approach requires optimizing a NeRF for each new target lighting condition. Furthermore, the computation of the latent NeRF involves a significant time investment (0.75 hours using 16 A100 GPUs). It would be beneficial to discuss potential solutions for this issue.

**Questions:**

- How many samples ($S$) were used during the experiments presented in Table 1 and Table 2?

- As this method models the probability of relit images and fixes $Z=0$ during inference, it would be interesting to investigate how the performance varies by sampling different $Z$ values.

- Typo in Line $446$

**Limitations:**

Limitations have been well discussed.

---

> ### Author Rebuttal · Authors · 2024-08-07
>
> ## About efficiency
> We acknowledge that our approach currently prioritizes quality over real-time performance, as generating new samples with the RDM and optimizing a NeRF for each new lighting condition is computationally intensive (L252-254).
>
> A potential solution to mitigate this limitation is to adopt an amortization strategy, similar to [a]. Specifically, we could train a single latent-NeRF conditioned on both latent codes and a latent representation of the given lighting. This design would allow us to train a single latent-NeRF model capable of handling multiple target illuminations. As a result, when presented with a new lighting condition, the trained model could directly generate the 3D relit version without requiring further optimization.
>
> [a] Lorraine et al., ATT3D: Amortized Text-to-3D Object Synthesis. ICCV 2023.
>
> ## Number of samples used for Tab. 1 and 2
>
> We use 16 samples per view for all benchmarking evaluations.
>
> ## Why is Z = 0?
>
> It is unclear how we could use the optimal Z that best matches the actual material besides optimizing Z using the test set images, which does not seem fair.  We found that setting Z to 0 yields good results across both our synthetic and real-world benchmarks (see Tab. 1 and 2, and Fig. 5 and 6). We'll clarify this in the final version of our paper.
>
> We provide more qualitative results for different latent codes in Fig. 1 (see PDF). These results demonstrate that the optimized latent codes effectively capture various plausible explanations of the materials.
>
>
> ## Typo
>
> Thanks a lot for pointing this out. We will correct it.

---

### Official Review · Reviewer_q4ex · 2024-07-10

**Soundness:** 3
**Presentation:** 4
**Contribution:** 2
**Rating:** 4
**Confidence:** 5

**Summary:**

The paper proposes a method for novel view synthesis with novel lighting given multi-view observation of an object. The pipeline is composed of a learned single-image-based relighting model based on image diffusion models, and a latent NeRF model that reconstructs the appearance of the relit object by reconciling relit images across views and samples. The relighting diffusion model (RDM) is conditioned on radiance cues generated from the geometry and target lighting under samples of roughness values, using a finetuned ControlNet architecture. The method is evaluated on both synthetic and real-world scenes, against baseline methods (mostly object-centric inverse rendering models), with both qualitative and quantitative results. Ablation study is also provided on design choices (e.g. latent/no latent NeRF, and number of diffusion samples).

**Strengths:**

[1] An alternative approach to object-centric relighting without optimization-based inverse rendering. The main novelty of the method which distinguishes it from previous methods of the task is, the method leverages a learned image-based diffusion model to directly predict the relit images, and then reconstructs 3D representation of the relit object for novel view synthesis. This approach opens up a new venue for solving the task besides baseline methods. The diffusion model is conditioned on 'radiance cues' as an input lighting proxy, which itself is not new in a learning-based inverse rendering setting, but the method is able to retarget the lighting proxy as condition for a diffusion model, which might inspire future works using diffusion models to hallucinate lighting.

[2] The design of the Latent NeRF to reconcile sampled relit results from the diffusion model. The paper demonstrates that the latent NeRF is able to better reconcile sampled results from a diffusion model while preserving the details and decent specular highlights.

[3] Extensive evaluation. The paper is able to include extensive evaluation on both real-world and synthetic datasets on various metrics.

**Weaknesses:**

[1] Effectiveness of the latent NeRF. The paper claims that the latent NeRF is able to reconcile difference in the sampled results from RDM. However, insufficient analysis is provided for justification. For instance, in Fig. 4, results in (a) seem to present more specularity than what is actually present in the hotdog scene (in other words, RDM seems to produce unrealistic relit images), and results from the latent NeRF in (b) seems to be an averaged version of the samples, with most of the specularity averaged out, arriving at a mostly diffuse appearance except for areas where specularity all agree among sampled. As a result, in Fig. 6, when compared to inverse rendering methods like Neural-PBIR, despite that the proposed method is able to produce better specularity in highlighted areas (thanks to explicitly conditioning on radiance cues), the results of the proposed method look 'dull', where shadows and high-frequency details seems to be lacking. This casts questions into the performance of both RDM to produce consistent and photorealistic relit images, as well as the ability of the latent NeRF to successfully reconcile inconsistency in the sampled results without sacrificing details and shadows.

[2] Claim on better secularity in the results is questionable. The paper explains that, when compared to Neural-PBIR the proposed method is in second place numerically because the approximate lighting ground truth from the light probe unfairly favors Neural-PBIR despite the proposed method produced more convincing specularity. However this is largely unjustified, as (1) it is not clear how much weight is on the specular pixels in the overall metrics (2) whether the proposed method actually produces better specularity in the absence of true ground truth of relit objects. Moreover as mentioned in [1], the proposed method seems to lose more details (e.g. in the spikes of the cactus, or details in the ball). In this case, even if the methods excels in specularity, this is not the single most important property that is accounted for in those metrics, or visually, when evaluating the quality of the results.

**Questions:**

[1] Quality of the sampled results from RDM. Why are some results look overly specular (e.g. the bread in the hotdog scene in Fig. 4). And if more examples can be provided on results from RDM?

[2] Reconciliation by the latent NeRF. Does the Latent NeRF cost details and specularity in sampled results from RDM when reconciling the results into a NeRF representation? Also if there are view-dependent effects in the sampled results, does the latent NeRF properly preserve those effects? Again, more examples are to be provided in order to further examine the behavior of the two modules beyond one single scene (and especially for synthetic scenes, where 'true' ground truth of relit objects is available). There are additional results on Standford-ORB scenes in the Supp but analysis is needed in the main paper to support the method using those results.

[3] Additional ablation. Is it helpful to use more or fewer roughness levels in generating radiance cues?

[4] In designing baselines, why not comparing against ALL methods in both synthetic and real-world settings?

---

> ### Author Rebuttal · Authors · 2024-08-07
>
> ## Fig. 4.(a)’s generations are unrealistic
> "More specularity than what is actually present in the hotdog scene" does not imply unrealistic results. Without known source lighting, the RDM samples from the entire distribution of plausible relighting outcomes. We showcased random samples from this distribution, some of which exhibit higher specularity on the hotdog. However, due to ambiguities between materials and lighting (L127), even these seemingly less-likely images are plausible. The subsequent latent-NeRF optimization process focuses on finding the most probable results.
> ## Averaged results in Fig. 4.(b)
> This outcome is to be expected: with a sufficiently large sample size, we accurately capture the underlying material distributions. Latent-NeRF maintains consistent appearance across all samples while mitigating inconsistencies. As shown in Fig. 4(b), it preserves the plate's specular effects seen in most samples from Fig. 4(a) while reducing the bread's glossiness, keeping mild specular effects on the sauces.
> ## Lack of detail in Fig. 6
> Our method surpasses all inverse rendering approaches in capturing high-frequency details, except for Neural-PBIR. Notably, for the salt can (column 2), our method clearly recovers the ingredient list, which most baselines miss. Similar improvements are evident for the toy car (column 6) and gnome (column 7).
>
> Compared to Neural-PBIR, our results often match or surpass its fine details, see the teapot (column 1), salt can (column 2), toy car (column 6), and gnome (column 7). The cactus (column 4) is an exception, due to UniSDF's inability to produce high geometric quality, as noted in the main paper (L251).
>
> Our method's results will inherently improve with advances in geometry reconstruction, as our RDM trains on perfect synthetic geometry. Neural-PBIR's code is unavailable, but its isolated geometry reconstruction stage (Sec. 3.1 in [39]) would allow us to directly leverage better geometry.
>
> ## Our results lacking shadows in Fig. 6
>
> Our method consistently produces shadows that accurately match the ground truth. Although Neural-PBIR shows more pronounced shadows on the toy car (column 6), these deviate significantly from the ground truth.
>
> Shadows depend on geometry and illumination, modeled through our radiance cues, leading to realistic shadow placement (Fig. 4.(a)). This accuracy is maintained through Latent-NeRF optimization (Fig. 4.(b)).
>
>
> ## Consistency of samples from RDM
>
> To clarify, we do not claim that our RDM generates "consistent" relit images. As our RDM performs single-image relighting, we anticipate diverse relit images across samples for the same view, as well as across different views (see Fig. 4.(a)). The role of the latent-NeRF is to reconcile these inconsistencies into a cohesive 3D relit result.
>
>
> ## Questioning claim that light probe unfairly favors Neural-PBIR
> We appreciate this valid concern and would like to clarify.
>
> Stanford-ORB provides estimated **per-image** illumination (L235 and [21]’s Fig. 4 and 5), moving a light probe for each image (PDF’s Fig. 2). Ideally, fixed illumination per object would match reality, but aligning the object and light probe is challenging. This limitation in the Stanford-ORB benchmark can significantly affect results, especially in areas with specular highlights (PDF’s Tab. 1). Consequently, there is no “correct” way to do relighting in Stanford-ORB: our results use fixed illumination, while competitors use per-image illumination.
>
> We rendered the ground truth geometry and materials, obtained by Stanford-ORB in a controlled studio environment ([21]’s Sec. 3.2.2). Each view was rendered under **fixed** illumination (consistent environment map) and **per-image** illumination (unique environment map per view). Fig. 3 in the PDF shows both renderings alongside the corresponding real image. Note the significant variations, especially in areas with specular highlights (see marked regions and PSNR).
>
> We computed metrics for each method using both reference renderings as ground truth (Tab. 1 in PDF). Our method excels under fixed illumination but performs worse with per-image illumination. Competitors show the opposite trend, doing better with per-image illumination.
>
> Neural-PBIR did not release code or full results, limiting comparisons to 8 images in [39]’s Fig. 10, where it also performs better with per-image setup. We outperform Neural-PBIR in most metrics:
> |8 images|PSNR-H $\uparrow$|PSNR-L $\uparrow$|SSIM$\uparrow$|LPIPS $\downarrow$|
> |----|-:|-:|-:|-:|
> |[fixed] Neural-PBIR |24.61|30.70|0.966|0.032|
> |[fixed] Ours |25.19|31.33|0.968|0.030|
> ||
> |[per-image] Neural-PBIR |25.03|30.97|0.966|0.032|
> |[per-image] Ours |25.04|30.76|0.967|0.030|
>
> Computational limitations prevented us from employing per-image illumination, as it would have required optimizing a latent-NeRF for each per-image illumination. Given the trends observed for competitors, we believe our method would demonstrate improved performance if this were feasible.
>
> ## More qualitative results for Fig. 4
>
> See Fig. 1 in the PDF.
>
> ## About view-dependent effects
>
> Our latent-NeRF can maintain view-dependent properties corresponding to each generation. Please refer to the cactus (4th row), pepsi (5th row), grogu (6th row), and teapot (7th row) scenes in Fig. 1 in the PDF.
>
> ## Ablation on #roughnesses
>
> We empirically found that using four radiance cues yielded strong results (Tab. 1 & 2, Fig. 5 & 6). Given the computational cost of training the RDM (L452), we did not conduct further ablations. However, we are happy to provide additional ablation studies in the final version if the reviewer would like us to.
>
> ## Why not compare against all baselines?
>
> We used the official benchmark results from [17] and [21] to ensure authenticity. We also obtained genuine qualitative results directly from the benchmark.
>
> The only exception is Neural-PBIR, as its code is not publicly available, preventing us from applying it to the TensoIR dataset.

---

> > ### Comment · Area_Chair_5NW5 · 2024-08-09
> > **Discussion**
> >
> > Dear reviewer q4ex
> >
> > We received diverse ratings for this submission, and you were the only one with a negative initial review. Please review the rebuttal and the comments from other reviewers and join the discussion as soon as possible. Thank you!
> >
> > Your AC

---

> > > ### Comment · Reviewer_q4ex · 2024-08-11
> > >
> > > I would like to thank the reviewers for their rebuttal to address my concerns. I agree with the responses to some of the minor concerns including ablation on roughness, and evaluation settings, but would insist to differ on the rest of the major issues. Specifically,
> > >
> > > [1] Unrealistic relit images from the hot dog scene in Fig. 4. The trained image diffusion model is supposed to make correct estimation on the material of the scene (and produce convincing images under novel lighting with correct materials true to the input images). However, assuming the input images are typical of the popular hot dog scene where the bun is mostly diffuse, some samples are unreasonably specular on those regions. I do agree with the author that there is inherent ambiguity between materials and lighting in inverse rendering, which sometimes lead to different plausible combinations of materials and lighting. However, (a) the diffusion model is supposed to make good estimation on the materials based on the input images, instead of producing results of diverse levels of specularity; (b) given multiview observation of the hot dog scene as input, where specularity is NEVER observed on any regions of the bun, there is minimal ambiguity between materials and lighting. There is little to no chance that the bun is actually of specular material while none of the multi-view input images suggest any specularity on the bun. In general, the ambiguity is more likely with single or few images, and more often in the case of baked shadows in materials, and less or not likely at all when there is sufficient observation (as is the setting with this paper) and no specularity is present in any view. In conclusion, I would incline to insist that the samples where the bun is specular are not plausible given the input images, and the trained diffusion model does not do a great job in estimating the true materials in some cases.
> > >
> > > Interestingly, in the additional 3 samples of the hot dog scene in the rebuttal PDF, the bun is not specular, while the bun is specular in sample #1, 2, 4, 5, of the 5 samples in Fig. 4(a). It is interesting to hear more from the authors on this difference to make a more informed conclusion on the behavior of the diffusion model.
> > >
> > > [2] On the benchmark with Standford-ORB. The authors included additional discussion on the capturing setting of the lighting GT with the dataset, indicating due to the light prob is not collocated with the object, the ground truth lighting is not accurate. I agree on this aspect but insist that due to this issue, the errors in lighting ground truth do not necessarily favor one method or the other. It is simply not reliable for ANY method. In conclusion, I would think it is not justified that the result from this method actually outperforms Neural-PBIR in the presence of the noisy GT lighting, and recommend the authors to rephrase the related language.
> > >
> > > [3] On the comparison of results against Neural-PBIR. I would conclude that the proposed method and Neural-PBIR both hit and miss in some aspects over the provided samples, w.r.t. level of details, correctness of specularity and shadows. None is overwhelmingly superior to the other. As a result, the performance of the proposed method is on par with Neural-PBIR over all without significant and meaningful improvement.
> > >
> > > Overall, I like the idea and the formulation of the method a lot, but believe there is no significant improvement in the results when compared with baselines. Drawbacks of the method and the paper are also pronounced: long and brittle pipeline (as suggested by Reviewer hgwb), modules (particularly RDM) not sufficiently evaluated, and unjustified claims on the performance. I would refrain from accepting the paper until hearing more from the authors and reviewers.

---

> ### Author Response · Authors · 2024-08-13
>
> We thank the reviewer for their time in reading our rebuttal and providing the feedback.
>
> We are pleased that the reviewer appreciates the concept and formulation of our method.
>
> Our primary objective was to introduce a novel paradigm with the potential to replace hand-crafted inverse rendering techniques. Although our current method may not significantly outperform all existing baselines, we believe it lays a strong foundation for future advancements in the field.
>
> We now address each question below:
>
> ## Response to Point [1]
>
> ### **Diffusion model input**
>
> There seems to be a misunderstanding with respect to the number of views used as input to our relighting diffusion model (RDM). The RDM model (L168 - 172)  is a **single-image/view** model. It has never been exposed to “multiview images” (see Eq. (4)).
>
> We also agree that it would be interesting to further improve the RDM model by conditioning on multiple views.
>
> ### **Unrealistic relit images | little to no chance that the bun is actually of specular material**
>
> We want to emphasize that our RDM models the entire distribution of plausible materials given a single observation of an object without known source lighting.
>
> While material roughness is still ambiguous when the source lighting is unknown, we agree that the samples with specular hot dog buns are less likely, and would hope that a future multi-view RDM model could improve this.
>
> ### **Different appearances for Hotdog’s bun between main paper and rebuttal**
>
> For Fig. 4(a) we chose to show a wide spectrum of plausible samples generated by the diffusion model.
>
> For the rebuttal we chose to show more samples, which included less specular buns.
>
> This further demonstrates the wide range of possible combinations between lighting and material. The final reconstructions recover the most likely materials (filter out the shiny bun samples) and do not exhibit any specular highlights for the bun, see Fig. 4(b).
>
> ## Response to Point [2]
>
> ### **On the benchmark with Stanford-ORB**
> We agree that in the presence of noisy GT lighting the analysis does not favor any method and will rephrase the related language.
>
> ## Response to Point [3]
>
> We show qualitatively that our method is the only one that somewhat-consistently recovers correct specularities on this dataset, while all other methods exhibit no specular highlights, apart from Neural-PBIR for a single example (the ball in the rightmost column of Figure 6). For the task of relighting this is perhaps one of the most important features.
>
> Unfortunately, Neural-PBIR only provides 8 qualitative result images (no source code is currently available), hence it is hard to properly compare our results with their approach, and we are happy to clarify our claims in the paper and limit them to only refer to specular highlights.
>
> We also admit that our method sometimes misses details due to the UniSDF reconstruction, and are happy to clarify this in a revised version.
>
>
> ## Overall
>
> We are happy that the reviewer recognizes the value of our proposed 3D relighting paradigm, which we consider the core contribution of our work (L42 and 256).
>
> Our method significantly outperforms all baselines except Neural-PBIR. However, due to limited results, unavailable code, and challenges with Stanford-ORB, it's difficult to determine which method is superior.
>
> We also agree that each component of our pipeline has the potential for further improvements. Since the components are independent of each other, improving one should improve overall performance.
>
> We are optimistic that future research will address these areas and build upon our framework.

---

### Official Review · Reviewer_hgwb · 2024-07-12

**Soundness:** 3
**Presentation:** 4
**Contribution:** 2
**Rating:** 5
**Confidence:** 4

**Summary:**

The paper proposes a method to relight NeRF representations from a set of multi-view input images. The paper first trains a NeRF model and recovers geometry using UniSDF from the input images. Then, from the mesh, they obtain radiance cues which are used to condition a diffusion model to sample relit images of the scene under a target lighting. Since these relit images are samples from a distribution of plausible relit images, the sampled results are used to train a latent code conditioned NeRF model, and the lighting of the scene can be manipulated by walking through the latent space. The core contribution of the paper is to use the relit images rather than trying to explicitly model and learn the lighting and material properties.

**Strengths:**

1. The paper shows that instead of recovering the material properties, it is enough to relight the training images and optimize the appearance to reach state-of-the-art results. The comparisons demonstrate that the results are state-of-the-art, which is important for improving the task of object relighting.
2. The diffusion model trained is described in detail and seems novel for the task of relighting. Details on the radiance cues and the architecture and training process of the diffusion model are described in detail. The model can be trained on synthetic data and applied to real-world data.
3. The lighting is multi-view consistent with the help of the latent-NeRF model
4. The presentation of the paper is clear, description of the method is high-quality, and comparisons are detailed

**Weaknesses:**

1. The significance of using diffusion models to relight images, which are then used to train a NeRF, is limited to the context of application to NeRFs. Concurrent works have very similar methods in the context of Gaussian splatting (https://repo-sam.inria.fr/fungraph/generative-radiance-field-relighting/) - what is the benefit of this approach over those? Some discussion should be made on this.
2. The pipeline seems brittle, requiring many steps to go right and limited by the sum of limitations of many different technologies. The quality of relighting is not only limited to the quality of the diffusion model output but also depends on the quality of the UniSDF mesh reconstruction, which is required to create the radiance cues. This means that a good capture is a necessary prerequisite to relight with this method, and one could simply relight the extracted mesh to achieve a similar effect.
3. I find the abstract a bit misleading: “we first relight each input image using an image diffusion model conditioned on lighting and then reconstruct a Neural Radiance Field (NeRF) with these relit images, from which we render novel views under the target lighting.” This description doesn’t accurately represent what actually happens, as it leaves out the most crucial part of the pipeline (reconstruction + conditioning) and instead suggests that the NeRF can be directly reconstructed from the relit images (Compare figure 2 of the paper).
4. The choice of UniSDF is neither justified nor are alternatives discussed. The part describing this aspect of the approach is relegated to the appendix. Consequently, there is no ablation study to determine if other reconstruction methods might work better.
5. With respect to figure 6: It makes sense to highlight the same area for all depicted approaches. For instance, the blocks and the ball have no highlights, but other approaches seem to perform better in these areas. Additionally, the normalization technique used to match the ground truth images is not ablated. Does this make a difference in the reconstructed quality? I notice that many of the other methods are darker than I’d expect and I wonder if this introduces some artifacts which hurt these methods and promote the proposed method.
6. It is unclear why the paper limits itself only to objects. UniSDF clearly shows good performance for scenes. The authors don’t discuss this limitation in the paper.
7. Changing the target light even slightly requires a wait of 45 minutes for a final rendering. Other approaches, as referenced above, seem to not require that much time.

**Questions:**

Many questions are discussed in the weaknesses, specifically:
1. How sensitive is the paper to the UniSDF reconstruction pipeline? Are there better methods which exist? Why does the paper only work for objects?
2. How does the normalization to ground-truth images for visualization of relit images actually affect the quality?

**Limitations:**

Some limitations are discussed in the paper, but there are additional ones which should be discussed in more detail, such as the long reconstruction time to get new lighting conditions.

---

> ### Author Rebuttal · Authors · 2024-08-07
>
> ## Method is limited to NeRF
>
> Our approach introduces a novel 3D relighting paradigm, employing a Relighting Diffusion Model (RDM) as prior to optimize a relit 3D representation. This approach is adaptable to various 3D representations like NeRFs or Gaussian Splats, since both can be conditioned on latent embeddings. For example, Wild Gaussians [a] (Sec 3.2) uses latent embeddings for a similar purpose in a different context (modeling lighting variations in internet photo collections with Gaussian Splatting).
>
> [a] Kulhanek et al., WildGaussians: 3D Gaussian Splatting in the Wild. ArXiv 2024
>
> ## Advantages over concurrent work
>
> 1. The concurrent work focuses on forward-facing indoor scenes, while ours addresses full 360 degree 3D object-centric relighting.
> 2. Our method can handle intricate environment map-based lighting, whereas the concurrent work is limited to single-source directional illumination, a subset of what IllumiNeRF supports.
> 3. The concurrent work’s diffusion model relights images only to 18 known light directions in the dataset (Sec. 3.2).
> 4. Their latent code is per-view, while ours is per-sample (See Eq. 3 and Fig. 8)
>
> Note that EGSR (July 3-5) took place after the NeurIPS submission deadline (May 22).
>
> ## The pipeline seems brittle
>
> Our presented numerical metrics and qualitative visualizations either produce SOTA results (Tab. 1 and Fig. 5) or highly compelling results (Tab. 2 and Fig. 6).
>
> While we acknowledge that our approach relies on high-quality geometry estimates from UniSDF (L249), it is crucial to note that our diffusion model's training relies solely on synthetic data. This design choice completely decouples the training process from the inference pipeline depicted in Fig. 2. Consequently, our method will benefit from future improvements in geometry reconstruction as well as improvements with regard to the relighting model or the latent NeRF optimization.
>
> ## Simply relighting the extracted mesh
>
> We are unsure exactly what the reviewer is suggesting.
>
> If the reviewer suggests simply relighting the output from UniSDF, it's important to emphasize that UniSDF generates only geometric information and outgoing radiance, not material properties. Therefore, direct relighting of the mesh is not possible.
>
> If the reviewer proposes to perform inverse rendering based on the geometry predicted by UniSDF, we would like to clarify that one of the main benefits of our method is that it does not require inverse rendering (L42 - 46).
>
>
> ## Abstract is misleading
>
> Thank you for the suggestion! We will modify the final version of the abstract to include reconstruction and conditioning.
>
> ## Use of UniSDF is not justified
>
> UniSDF is a SOTA method for surface/SDF-based NeRF approaches (L387) on the Shiny Blender benchmark [43] (see MAE in [47]’s Tab. 1). We build on top of it, as other non-surface methods would make it harder to obtain high-quality geometry for the radiance cues computation (L249, 388 - 390). Our framework can easily be applied to any NeRF/SDF approaches that recover better geometry in the future.
>
> ## About highlights in Fig. 6
>
> We will integrate the changes as suggested.
>
> ## About normalization for Fig. 6
>
> The qualitative baseline results were kindly provided by the authors of Stanford-ORB [21]. Additionally, we utilized the benchmark's official code (https://github.com/StanfordORB/Stanford-ORB/blob/962ea6d2cc/scripts/test.py/#L36) to perform the normalization.
>
> ## Why focus on objects?
>
> Good question. There are multiple reasons for focusing on objects:
> 1. Object-centric 3D relighting is vital in various downstream applications, e.g., AR/VR, robotics, and game development.
> 2. Scene-centric lighting is a much harder task to solve. It is hard to set up realistic illumination for scenes. Conditioning on lighting is non-trivial as lighting representations such as environment maps are not easily applicable to scenes.
> 3. Established synthetic and real-world object-centric benchmarks like TensoIR and Stanford-ORB enable clear evaluation and comparison.
> 4. Large-scale object-centric synthetic datasets like Objaverse [10] are readily available for training the relighting diffusion model.
>
> ## Efficiency of method
> We acknowledge that our approach currently prioritizes quality over real-time performance, as generating new samples with the RDM and optimizing a NeRF for each new lighting condition is computationally intensive (L252-254).
>
> However, we believe this focus on quality is crucial as a first step. Our extensive evaluations and thorough numerical experiments demonstrate competitive results, offering a "promising alternative for relighting" (mZi1). We hope our work "inspires future works using diffusion models to hallucinate lighting" (q4ex), with a gradual focus on improving efficiency.
>
> We see parallels in the NeRF community, where the initial MLP-based NeRF [30], despite its impressive results, was computationally demanding. Subsequent research, like instant-NGP [31] and Gaussian Splatting, significantly improved efficiency. We envision a similar trajectory for our approach.

---

> > ### Comment · Reviewer_hgwb · 2024-08-13
> >
> > Thank you for the detailed rebuttal. It has addressed the majority of my concerns, and I do see improvement over prior work in terms of quality and acknowledge the novel contributions. However, I do feel as though the computational drawback and reliance on geometry reconstruction limit the applicability of the method in its current state enough such that it might not be presenting significant or impactful contributions at this time. I remain slightly positive on the paper.

---

> ### Author Response · Authors · 2024-08-13
>
> We thank the reviewer for their time in reading our rebuttal and providing feedback.
>
> We are encouraged that the reviewer recognizes the quality improvements and novel contributions introduced by our method.
>
> We would like to clarify that every existing 3D relighting baseline also requires geometry reconstruction.
>
> Our current method did not focus on improving speed, but rather introducing a novel 3D relighting paradigm. It was designed as a general framework that can be further developed to substantially improve speed in the future, e.g. by using Gaussian Splatting representation instead of NeRF as well as using a faster diffusion sampler.

---

### Official Review · Reviewer_Ma7D · 2024-07-13

**Soundness:** 3
**Presentation:** 4
**Contribution:** 3
**Rating:** 5
**Confidence:** 4

**Summary:**

The paper proposes a new method called illumiNeRF for 3D relighting — given a set of posed input images under an unknown lighting, illumiNeRF produces novel views relit under a target lighting. Most of existing methods use inverse rendering to first recover the material and lighting in the given images, and apply new lighting to the object. The authors point out there is inherent ambiguity in decomposing materials and lighting. Instead, they propose a probabilistic model of representing multiple possible materials as latents, and train a latent NeRF using a relighting diffusion model. The authors compare their method with existing baselines and outperform most of them.

**Strengths:**

- The problem formulation in sec 3.1 is explained clearly. The authors point out the inherent ambiguity in inverse rendering problems when decomposing materials and lighting.
- The probabilistic model of representing multiple possible materials as latents in NeRF is very interesting.

**Weaknesses:**

- Although during training, multiple materials are considered as different Z when optimizing the latent-NeRF, Z is manually set to 0 at test time, which corresponds to a specific explanation. However, Z=0 does not seem to represent the most likely material because the authors assume a uniform prior over Z according to line 163.
- From Figure 6, the results lose a lot of details compared to other baselines like Neural-PBIR. For example, the method lost details of the spikes in the cactus sample. In table 2, Neural-PBIR also outperforms the proposed method.

**Questions:**

It would be great to show 3D renderings with different latent explanations in Figure 4(b).

**Limitations:**

Yes

---

> ### Author Rebuttal · Authors · 2024-08-07
>
> ## Why is Z = 0?
>
> It is unclear how we could use the optimal Z that best matches the actual material besides optimizing Z using the test set images, which does not seem fair.  We found that setting Z to 0 yields good results across both our synthetic and real-world benchmarks (see Tab. 1 and 2, and Fig. 5 and 6). We'll clarify this in the final version of our paper.
>
> We provide more qualitative results for different latent codes in Fig. 1 (see PDF). These results demonstrate that the optimized latent codes effectively capture various plausible explanations of the materials.
>
> ## Result quality in Fig. 6
>
> Our method consistently surpasses all inverse rendering approaches in capturing high-frequency details, except for Neural-PBIR. Notably, for the salt can (column 2), our method clearly recovers the ingredient list, which most baselines miss. Similar improvements are evident for the toy car (column 6) and gnome (column 7).
>
> Compared to Neural-PBIR, our results often match or surpass its fine details, see the teapot (column 1), salt can (column 2), toy car (column 6), and gnome (column 7). The cactus (column 4) is an exception, due to UniSDF's inability to produce high quality geometry, as noted in the main paper (L251).
>
> Our method's results will inherently improve with advances in geometry reconstruction, as our RDM is trained on perfect synthetic geometry. Neural-PBIR's code is unavailable, but its isolated geometry reconstruction stage (Sec. 3.1 in [39]) would allow us to directly leverage better geometry.
>
> ### About more examples for Fig. 4 (b)
>
> We provided more examples in Fig. 1 of the rebuttal PDF. We are happy to include these in a final version.

---

### Author Rebuttal · Authors · 2024-08-07

We thank all reviewers for their time and effort spent reviewing the paper.

We are grateful to hear that the reviewers agree that our work “addresses a long-standing challenge by introducing a novel paradigm" (mZi1).

We also appreciate the praise for our "probabilistic model of representing multiple possible materials as latents in NeRF" (Ma7D), the recognition of our work's importance for "improving the task of object relighting" (hgwb), and the potential to "inspire future works using diffusion models to hallucinate lighting" (q4ex).

Furthermore, we are pleased reviewers recognize our work’s “extensive evaluation" (q4ex) and find "the results are convincing" (mZi1).

We are also delighted that all reviewers gave the highest rating for the presentation, describing it as "clear" (hgwb), "well-structured" (mZi1), and "explained clearly" (Ma7D).

Below, we address each reviewer's questions.

---

### Decision · Program_Chairs · 2024-09-25

**Decision:**

Accept (poster)

**Comment:**

All reviewers agree that the idea of leveraging a learned image-based diffusion model to predict relit images for latent NeRF optimization and relighting is novel and interesting. The method also demonstrated superior rendering quality compared to prior inverse rendering methods (except Neural-PBIR) on the TensoIR and Stanford-ORB benchmarks. However, the reviewers also raised concerns about the effectiveness of the learned latent NeRF, noting that the diffusion model seems unreliable compared to dense view conditioning for material estimation, and the overall pipeline appears brittle. AC believes that the diffusion-based solution is fresh and valuable to the community and thus recommends acceptance.